# Unifying VLM-Guided Flow Matching and Spectral Anomaly Detection for Interpretable Veterinary Diagnosis

## Abstract

Automatic diagnosis of canine pneumothorax is challenged by data scarcity and the need for trustworthy models. To address this, we first introduce a public, pixel-level annotated dataset to facilitate research. We then propose a novel diagnostic paradigm that reframes the task as a synergistic process of signal localization and spectral detection. For localization, our method employs a Vision-Language Model (VLM) to guide an iterative Flow Matching process, which progressively refines segmentation masks to achieve superior boundary accuracy. For detection, the resulting mask is used to isolate features from the suspected lesion. We then apply Random Matrix Theory (RMT), a departure from traditional classifiers, to analyze these features. This approach models healthy tissue as predictable random noise and identifies pneumothorax by detecting statistically significant outlier eigenvalues that represent a non-random pathological signal. The high-fidelity localization from Flow Matching is crucial for purifying the signal, thus maximizing the sensitivity of our RMT detector. This synergy of generative segmentation and first-principles statistical analysis yields a highly accurate and interpretable diagnostic system.

## 1 Introduction

Canine pneumothorax is a common and potentially life-threatening emergency in veterinary clinical practice characterized by abnormal accumulation of gas in the pleural space between the lungs and the chest wall, resulting in lung collapse and severe respiratory distress Dickson et al. (2021); Jobson (2016). Timely and accurate diagnosis is essential to guide emergency treatment and improve prognosis. At present, chest X-ray radiography is a common method for the diagnosis of canine pneumothorax. However, the interpretation of radiological images is highly dependent on the expertise and clinical experience of veterinarians. In some subtle or atypical cases, manual interpretation may be subjective, and in emergency situations, it is challenging to quickly and accurately delineate the extent of collapse for assessing the severity of the disease and making treatment plans (such as thoracocenesis). Therefore, it is of great clinical application value to develop an intelligent tool that can assist veterinarians in rapid, objective and accurate diagnosis.

Recently, artificial intelligence technology represented by deep learning has made breakthroughs in the field of medical image analysis, and shows great potential especially in lesion segmentation and classification tasks Azad et al. (2024); Asgari Taghanaki et al. (2021); Antonelli et al. (2022). In veterinary radiology, AI algorithms have been initially applied to tasks such as assessment of canine hip dysplasia Loureiro et al. (2025), heart size measurement Ramisetty (2024), and identification of certain skeletal abnormalities Kostenko et al. (2024), showing great potential for improving diagnostic objectivity and efficiency. However, these traditional AI methods face two major bottlenecks. One is the extreme scarcity of large-scale, high-quality labeled data. There is a serious lack of standardized public datasets with high-quality expert annotations in the field of veterinary imaging. The construction of such a dataset is not only costly, but also requires the time of a large number of veterinary radiology experts. The second is the lack of interpretability. As illustrated in Figure 1, traditional models often function as "black boxes" that usually only provide numerical results for segmentation or classification and are unable to explain their diagnostic rationale, which limits their application in clinical decision making where a high degree of trust is required. In

contrast, our proposed framework provides a transparent and trustworthy alternative by combining precise lesion localization with a quantitative anomaly score, which is critical for clinical decision making. With the development of large-scale pre-trained Foundation Models, especially large language models (LLMS) and Vision-language models (VLMS) Touvron et al. (2023); Zhang et al. (2024), these models have gained unprecedented world knowledge and powerful zero-shot/few-shot inference capabilities through pre-training on massive multi-modal data. Their unique ability to understand and generate natural language opens up entirely new possibilities for building trustworthy human-computer interactive diagnostic systems Rane et al. (2023). Although LLM has shown great potential in the field of general human medicine, there is still a huge research gap in the highly specialized field of veterinary radiology.

To address the data scarcity problem, We begin by constructing and releasing the first publicly available radiological image dataset containing pixel-level expert annotations for canine pneumothorax. Based on this foundation, we propose an innovative VLM-FlowMatch segmentation framework, semantically guided lesion localization by iteratively refining an initial segmentation mask with a VLM-guided vector field. Finally, for the diagnostic task itself, we introduce a novel paradigm based on Random Matrix Theory (RMT) for anomaly detection, which quantifies the statistical perturbation from pathological signals within the focused lesion area to provide a robust Spectral Anomaly Score (SAS).

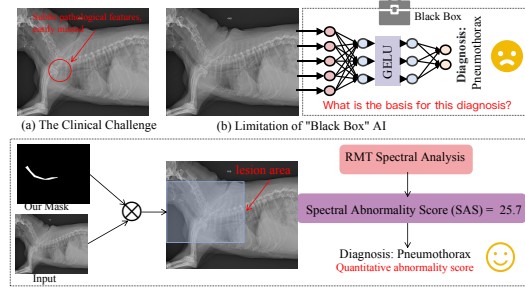

Figure 1: Comparison of diagnostic approaches for canine pneumothorax. (a) The clinical challenge of subtle features. (b) The interpretability issue of "black box" AI. (c) Our proposed framework with precise les

## 2 RELATED WORK

**Canine medical image segmentation.** Medical image segmentation is the cornerstone of computer-aided diagnosis, which aims to accurately identify anatomical structures and lesion regions at the pixel level Cui et al. (2023). Fully supervised deep learning models, represented by U-Net and its variants, have achieved outstanding achievements in numerous segmentation tasks and become the gold standard in this field Ronneberger et al. (2015); Cao et al. (2022). However, the success of these models is premised on large-scale, high-quality pixel-level labeled data. In specialized fields such as veterinary radiology, the cost of obtaining such data is extremely high, severely limiting the application of fully supervised methods Xiao et al. (2025). To address this challenge, the research community has explored a variety of data-efficient learning strategies, aiming to learn more robust features from limited labeled data. These methods include transfer learning Kim et al. (2022), weakly supervised learning Ren et al. (2023), and advanced techniques based on feature matching and distribution alignment Huang et al. (2024). UnetFlowMatch adopted in our work strengthens the model's understanding of the intrinsic structure of images through a novel matching mechanism Wang et al. (2025a). Although these data-efficient methods effectively alleviate the problem of data dependence, the trained models still face two major limitations. One is the accuracy bottleneck when dealing with fuzzy and subtle boundaries. The second is the inability to provide a credible explanation for the diagnosis.

**Applications of Large Language Models in Medical Imaging.** In recent years, large language models (LLMS) and vision-language models (VLM) have brought advances to the field of medical image analysis Wang et al. (2024a); Fang et al. (2024). Although traditional deep learning models perform well on tasks such as classification or segmentation, their nature of not being able to communicate effectively with clinicians has been a major obstacle in their clinical translation. LLM has advanced logical reasoning and natural language interaction capabilities, which can transform complex pixel information into language that human doctors can understand and verify Li et al. (2024). In Visual Question answering (VQA) and diagnostic AIDS, models are able to respond to natural language questions (such as Are there abnormalities in the image? ) to answer the specific content of the image, and even directly give preliminary diagnosis and classification recommendations Bazi et al. (2023). In the automatic generation of radiology reports, the model automatically analyzes the input medical images and generates a structured and standardized diagnostic report, which can

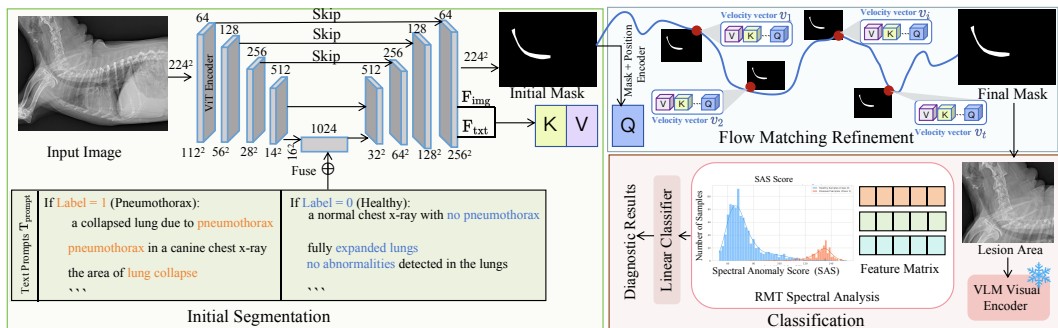

Figure 2: Overview of our proposed synergistic framework for canine pneumothorax diagnosis.

reduce the work burden of radiologists and standardize the quality of the report Alfarghaly et al. (2021); Wang et al. (2025b). These applications fully demonstrate the powerful ability of LLM to receive a processed image and output the final cognitive result. However, the reliability and factual accuracy of the model are still huge challenges, and sometimes it will produce plausible but inconsistent illusion Scirè et al. (2024). Most studies use LLM as an isolated, end-process module that lacks intervention and insight into upstream image processing steps such as segmentation.

## 3 METHOD

To achieve high-precision and semantically coherent segmentation of canine pneumothorax, our framework VLM-FlowMatch, reframes diagnosis as a unified process of signal localization and spectral analysis. As shown in Figure 2, it first employs a VLM-Infused U-Net and an Attentional Flow Matching module to generate a high-precision segmentation mask $\hat{M}$. This mask then serves as a crucial spatial filter to isolate the region of interest, enhancing the signal-to-noise ratio for our subsequent analysis. Finally, features from this focused region are fed into a Random Matrix Theory (RMT) based classifier to quantify their statistical deviation from a healthy baseline and render a final diagnosis.

### 3.1 ViT-UNet FOR INITIAL MASK GENERATION

The foundation of our model is a U-Net architecture where the entire encoder-decoder feature pathway is driven by a single pre-trained Vision Transformer (ViT). This design leverages the ViT's powerful global feature extraction capabilities for both semantic understanding in its final layers and providing multi-scale spatial details for the skip connections. Given an input image X, the ViT visual encoder processes it into a final visual feature map, which is then fused via element-wise multiplication with the projected feature vector from a text prompt $T_{prompt}$ to infuse semantic guidance.

A key innovation of our architecture lies in how the skip connections are generated. Instead of using a separate CNN encoder, we derive all skip-connection features directly from the ViT's final visual feature map. This map is progressively upsampled via bilinear interpolation to match the spatial resolutions of the decoder's different stages. The standard U-Net decoder then takes the text-fused visual features and this hierarchy of ViT-derived skip connections to reconstruct the initial segmentation mask $M^{(0)}$. The entire ViT-UNet model, denoted as $\Psi$, can be summarized as:

$$M^{(0)}, F_{img}, F_{txt} = \Psi(X, T_{prompt}; \theta_{vlm\text{-}unet}) \tag{1}$$

where $F_{img}$ represents the original visual features and $F_{txt}$ represents the text feature vector, both of which are passed to the subsequent refinement stage, $\theta$ represents the learnable parameters.

### 3.2 ITERATIVE REFINEMENT VIA VLM-GUIDED FLOW MATCHING

To further enhance the segmentation accuracy of the initial mask $M^{(0)}$, we learn a vector field $v$ through flow matching under the guidance of the rich features of VLM. During this process, the direction of the flow is guided by the VLM features at each step.

Our key innovation is how we predict this vector field. At each time step $t$ of the iterative process, the current segmentation state $x_t$ is used to form a *query* for a cross-attention mechanism. The *key*

and *value* are constructed by concatenating the VLM's text features $F_{txt}$ and image patch features $F_{img}$. This allows the model to ask, Given my current segmentation state, where should I adjust the boundaries based on the visual evidence and the textual description of pneumothorax?

Let the attentional flow module be $\Phi_{flow}$. The refinement process is an iterative update, which can be seen as a discretization of an ordinary differential equation (ODE):

$$x_{t+dt} = x_t + v(x_t, F_{img}, F_{txt}) \cdot dt \tag{2}$$

Where the velocity vector $v$ is predicted by the network, which internally computes:

$$v_t = \Phi_{flow}(\text{CrossAttention}(Q = f(x_t), K = [F_{txt}; F_{img}], V = [F_{txt}; F_{img}])) \tag{3}$$

Here, $f(x_t)$ represents the features extracted from the current mask state $x_t$. This process is repeated for $T$ steps, starting from $x_0 = M^{(0)}$, to yield the final, high-precision mask $\hat{M} = x_T$. This deep integration of VLM features at every step provides continuous, fine-grained semantic guidance.

We leverage the final segmentation mask $\hat{M}$ to isolate the region of interest. Specifically, we perform an element-wise multiplication (Hadamard product) between the original color image X and the binary mask $\hat{M}$, defined as:

$$X_{focus} = X \odot \hat{M} \tag{4}$$

where all pixels corresponding to irrelevant background and healthy tissue are zeroed out, effectively focusing the subsequent analysis solely on the potential pneumothorax area. By eliminating the statistical "noise" from non-pathological regions, the feature matrix extracted from $X_{focus}$ provides a much cleaner representation of the abnormality. This makes the underlying non-random "signal" of the pathology significantly more prominent and detectable.

### 3.3 FOCUSED DIAGNOSTIC CLASSIFICATION VIA SPECTRAL ANOMALY DETECTION

Although a healthy X-ray of the lungs has complex image content, after being mapped to a high-dimensional feature space by a visual language model (VLM), the statistical relationships (such as correlations) among its features follow a complex but predictable distribution, similar to a high-dimensional random system. When pneumothorax collapse occurs in the lungs, this structured lesion introduces a non-random signal in the feature space. Traditional methods (such as CNN) focus on learning the spatial shape of the signal itself, for example, the shape of the lung margin line. However, our method uses the powerful mathematical tool Random Matrix Theory (RMT) to detect the dramatic changes in the statistical characteristics of the entire system caused by the signal. Thus, the existence of anomalies can be determined, and the diagnostic task is defined from the traditional classification problem to the noise detection problem.

**Step 1: The Null Hypothesis $H_0$ and Empirical Spectral Distribution.** Our $H_0$ is that the features of a healthy lung region behave as high-dimensional noise. We model the VLM's output for a healthy, focused image $X_{focus}$ as a feature matrix $F_p \in \mathbb{R}^{N \times p}$, whose entries are independent and identically distributed random variables with zero mean and variance $\sigma^2 = 1$.

We analyze the spectrum of the sample covariance matrix $S = \frac{1}{N}F_p^T F_p$. The distribution of its eigenvalues $\{\lambda_i\}_{i=1}^p$ can be described by the **Empirical Spectral Distribution (ESD)**, defined as:

$$\mu_S(x) = \frac{1}{p}\sum_{i=1}^{p}\delta(x - \lambda_i) \tag{5}$$

where $\delta(\cdot)$ is the Dirac delta function. The ESD is essentially a histogram of the eigenvalues.

**Step 2: Spectral Analysis using Random Matrix Theory** The Marchenko-Pastur (MP) law states that as the matrix dimensions approach infinity ($N, p \to \infty$) such that the aspect ratio $p/N \to y \in (0, \infty)$, the ESD $\mu_S(x)$ converges to a deterministic probability distribution $f_{MP}(x)$. This **Marchenko-Pastur distribution** is the theoretical spectrum for random noise. Its probability density function (PDF) is given by:

$$f_{MP}(x) = \frac{1}{2\pi\sigma^2 yx}\sqrt{(\lambda_+ - x)(x - \lambda_-)} \tag{6}$$

where $x \in [\lambda_-, \lambda_+]$, and 0 otherwise. The support of this distribution is bounded by $\lambda_\pm = \sigma^2(1 \pm \sqrt{y})^2$. Any eigenvalue of our computed covariance matrix $S$ that exceeds this theoretical

maximum, $\lambda > \lambda_+$, is considered an outlier eigenvalue. These outliers are treated as the signature of a strong, non-random signal embedded within the features, corresponding to the pathological pattern of pneumothorax. Under $H_0$, all eigenvalues of $S$ should fall within this continuous spectrum.

**Step 3: The Spiked Covariance Model for Anomaly Detection**    Our alternative hypothesis ($H_1$), corresponding to a diseased lung, is that the feature matrix is not pure noise, but a sum of a random noise matrix $W$ and a low-rank, non-random signal matrix $U$. This signal represents the structured pathological information of pneumothorax.

$$\mathrm{F}_p = W + U, \quad \text{where } \mathrm{rank}(U) \ll p \tag{7}$$

This leads to a **"spiked" covariance model**. This model precisely describes the phenomenon that when a low-rank signal is added to a pure noise matrix, one or more eigenvalues of its covariance matrix will significantly deviate from the principal spectrum, forming isolated "spikes". The covariance matrix of $\mathrm{F}_p$ will have most of its eigenvalues conforming to the Marchenko-Pastur distribution (from the noise $W$), but one or more eigenvalues will "spike" out and lie beyond the upper edge $\lambda_+$. These **outlier eigenvalues** are the mathematical manifestation of the disease signal.

To quantify this signal, we define the **Spectral Anomaly Score (SAS)** as the total energy of these spikes relative to the noise edge:

$$\mathrm{SAS}(\mathrm{X}_{\text{focus}}) = \sum_{\lambda_i > \lambda_+} (\lambda_i - \lambda_+) \tag{8}$$

Finally, this scalar SAS value, which captures the strength of the pathological signal, is used as the sole feature for a linear classifier $\Psi_{\text{clf}}$ to make the diagnosis $\hat{Y} = \Psi_{\text{clf}}(\mathrm{SAS})$.

### 3.4 Optimization Objective

Our framework employs a staged training strategy, where key components are optimized independently with their specific objective functions.

**Segmentation Model Training:** The initial segmentation model $\Phi_{\text{seg}}$ and the refinement network $\Phi_{\text{refine}}$ are trained to minimize the discrepancy between the predicted mask and the ground truth. We use a hybrid loss function, $\mathcal{L}_{\text{seg}}$, composed of the Dice loss and Binary Cross-Entropy (BCE) loss to optimize for both regional overlap and pixel-wise accuracy.

$$\mathcal{L}_{\text{seg}}(M_{\text{pred}}, M_{gt}) = \mathcal{L}_{\text{Dice}}(M_{\text{pred}}, M_{gt}) + \lambda_{\text{bce}} \mathcal{L}_{\text{BCE}}(M_{\text{pred}}, M_{gt}) \tag{9}$$

where $M_{\text{pred}}$ is the predicted mask, $M_{gt}$ is the ground truth mask, and $\lambda_{\text{bce}}$ is a balancing hyperparameter.

**Diagnostic Classifier Training:** The objective of the diagnostic classifier $\Psi_{\text{diag}}$ is to accurately predict the presence of pneumothorax. Acknowledging the common issue of class imbalance in medical datasets, we fine-tune this model using the **Focal Loss**, $\mathcal{L}_{\text{Focal}}$. By adding a modulating factor, the Focal Loss dynamically down-weights the contribution of well-classified examples, forcing the model to focus on hard-to-classify samples. It is defined as:

$$\mathcal{L}_{\text{Focal}}(p_t) = -\alpha_t (1 - p_t)^\gamma \log(p_t) \tag{10}$$

where $p_t$ is the model's estimated probability for the ground truth class, $\gamma \geq 0$ is the focusing parameter that adjusts the rate at which easy examples are down-weighted, and $\alpha_t$ is a weighting factor to balance class importance.

## 4 Experiments

### 4.1 Dataset

The dataset used in our study was sourced from the public Canine Thoracic Radiograph collection available on the Korean AI-Hub platform (https://aihub.or.kr/). To guarantee the reproducibility of our research, we partitioned this curated dataset into fixed training, validation, and test sets. Specifically, the training set comprises 8641 images, the validation set comprises 2468 images, and the test set comprises 1236 images. All model training and evaluation reported in this paper were conducted on this fixed partition to ensure fair and comparable results.

## 4.2 EVALUATION METRICS

To comprehensively evaluate the performance of our framework, we adopt standard evaluation metrics for both segmentation and classification tasks. For segmentation performance, we use two widely accepted metrics to measure the agreement between the predicted mask $\hat{M}$ and the ground truth mask $M_{gt}$: the Dice Similarity Coefficient (mDice) and the mean Intersection over Union (mIoU). For the final diagnosis of pneumothorax collapse, we report Accuracy, Precision, Recall, and F1-Score.

## 4.3 IMPLEMENTATION DETAILS

Our framework was implemented using PyTorch. The UnetFlowMatch model was trained on our training set using the Adam optimizer with an initial learning rate of 1e-4. We utilized openclip as the evaluation and feedback model. The prompts for the VLM were carefully designed to elicit structured refinement instructions. The same VLM was used for diagnostic classification. All experiments were conducted on an NVIDIA 4090 GPU (48 GB).

## 4.4 QUANTITATIVE RESULTS

### 4.4.1 COMPARISON ON SEGMENTATION PERFORMANCE

We compare our method with several mainstream segmentation models Ronneberger et al. (2015); Zhou et al. (2018); Wang et al. (2025a); Qin et al. (2020); Cao et al. (2022); Wang et al. (2020); Badrinarayanan et al. (2017); Diakogiannis et al. (2020); Kirillov et al. (2023); Chen et al. (2018); Liu et al. (2021); Wang et al. (2024b); Liu et al. (2024). Table 1 presents a comprehensive performance comparison between our method and a variety of state-of-the-art segmentation models. Our model consistently ranks first across all metrics on both the validation and test sets. Specifically, on the test set, our method achieves a top mDice of 0.8953 and mIoU of 0.8114. This performance not only surpasses classic U-Net-based architectures like PolypFlow (0.8019 mIoU) and powerful Transformer-based models like DeepLabv3+ (0.7733 mIoU), but also significantly outperforms other recent Mamba-based approaches such as Swin-UMamba (0.7820 mIoU). The consistent lead on both validation and test sets also suggests a strong generalization ability of our model. These results robustly validate the superiority of our proposed framework with its VLM-guided module.

Table 1: Performance comparison with state-of-the-art segmentation methods.

| Category | Year | Model | Validation Set | | Test Set | |
|---|---|---|---|---|---|---|
| | | | mDice↑ | mIoU↑ | mDice↑ | mIoU↑ |
| Unet-based | 2015 | Unet | 0.8830 | 0.7949 | 0.8774 | 0.7878 |
| | 2018 | Unet++ | 0.8724 | 0.7811 | 0.8712 | 0.7788 |
| | 2025 | PolypFlow | 0.8917 | 0.8084 | 0.8869 | 0.8019 |
| | 2020 | $U^2$Net | 0.8890 | 0.8044 | 0.8834 | 0.7965 |
| | 2022 | Swin-UNet | 0.8559 | 0.7547 | 0.8462 | 0.7424 |
| Others | 2020 | HRNet | 0.8849 | 0.7978 | 0.8780 | 0.7883 |
| | 2017 | SegNet | 0.8825 | 0.7955 | 0.8777 | 0.7880 |
| | 2020 | ResUnet | 0.8725 | 0.7807 | 0.8670 | 0.7727 |
| | 2023 | SAM | 0.6710 | 0.5251 | 0.6731 | 0.5277 |
| Transformer-based | 2018 | DeepLabv3+ | 0.8740 | 0.7822 | 0.8681 | 0.7733 |
| | 2021 | Swin-Transformer | 0.5838 | 0.4224 | 0.5763 | 0.4165 |
| Mamba-based | 2024 | Mamba-UNet | 0.8566 | 0.7564 | 0.8506 | 0.7481 |
| | 2024 | Swin-UMamba | 0.8794 | 0.7906 | 0.8733 | 0.7820 |
| | | Ours | **0.9104** | **0.8217** | **0.8953** | **0.8114** |

### 4.4.2 COMPARISON ON DIAGNOSTIC CLASSIFICATION PERFORMANCE

We evaluated our framework on the diagnostic classification task against a comprehensive suite of baseline models Szegedy et al. (2015); Simonyan & Zisserman (2014); He et al. (2016); Huang et al. (2017); Szegedy et al. (2016); Chollet (2017); Szegedy et al. (2017); Zoph et al. (2018); Tan & Le (2019); Dosovitskiy et al. (2020); Wu et al. (2021); Bao et al. (2021), as summarized in Table 2. A key challenge of our dataset is class imbalance, making the F1-score the primary metric for robust

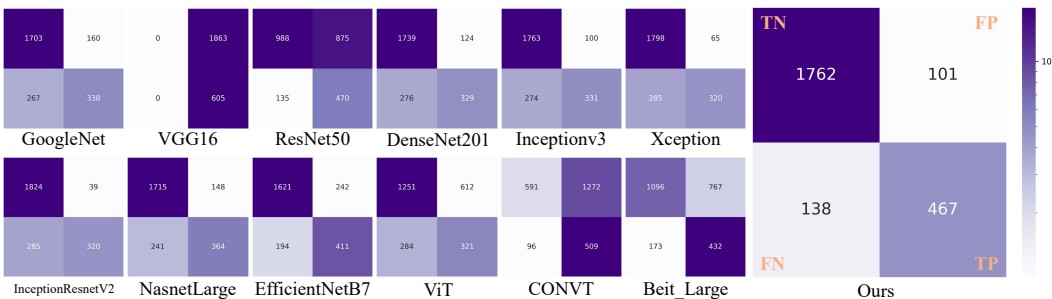

Figure 3: This figure presents a comparative analysis of the confusion matrices for the proposed model and twelve other models. Each matrix displays the counts for True Negatives (TN) , False Positives (FP) , False Negatives (FN) , and True Positives (TP). The results highlight the superior performance of our model, which achieves a strong balance in correctly identifying both positive and negative instances while maintaining low error rates compared to the other methods.

evaluation. The results clearly highlight the superiority of our proposed method. On the test set, our model achieves the highest accuracy of 0.9032 and, more importantly, the highest F1-score of 0.7962.

Table 2: Performance comparison on the validation and test sets.

| Model | Validation Set | | | | Test Set | | | |
|---|---|---|---|---|---|---|---|---|
| | Acc.↑ | Prec.↑ | Rec.↑ | F1↑ | Acc.↑ | Prec.↑ | Rec.↑ | F1↑ |
| GoogleNet | 0.8576 | 0.6176 | 0.6336 | 0.6255 | 0.8270 | 0.6787 | 0.5587 | 0.6129 |
| VGG16 | 0.1877 | 0.1877 | **1.0000** | 0.3161 | 0.2451 | 0.2451 | **1.0000** | 0.3938 |
| ResNet50 | 0.5793 | 0.2889 | 0.8491 | 0.4311 | 0.5908 | 0.3494 | 0.7769 | 0.4821 |
| DenseNet201 | 0.8835 | 0.7222 | 0.6164 | 0.6651 | 0.8379 | 0.7263 | 0.5438 | 0.6219 |
| Inceptionv3 | 0.8875 | 0.7246 | 0.6466 | 0.6834 | 0.8485 | 0.7680 | 0.5471 | 0.6390 |
| Xception | 0.9110 | **0.8631** | 0.6250 | 0.7250 | 0.8582 | 0.8312 | 0.5289 | 0.6465 |
| InceptionResnetV2 | 0.9100 | 0.8588 | 0.6293 | 0.7264 | 0.8687 | **0.8914** | 0.5289 | 0.6639 |
| NasnetLarge | 0.8827 | 0.6933 | 0.6724 | 0.6827 | 0.8424 | 0.7109 | 0.6017 | 0.6517 |
| EfficientNetB7 | 0.8592 | 0.5993 | 0.7543 | 0.6679 | 0.8233 | 0.6294 | 0.6793 | 0.6534 |
| Vision Transformer | 0.6286 | 0.2725 | 0.5862 | 0.3721 | 0.6370 | 0.3441 | 0.5306 | 0.4174 |
| CONVT | 0.4142 | 0.2248 | 0.8664 | 0.3570 | 0.4457 | 0.2858 | 0.8413 | 0.4267 |
| Beit_large | 0.6052 | 0.2621 | 0.6078 | 0.3662 | 0.6191 | 0.3603 | 0.7140 | 0.4789 |
| Ours | **0.9126** | 0.7583 | 0.7845 | **0.7712** | **0.9032** | 0.8222 | 0.7719 | **0.7962** |

VGG16 and CONVT show high recall but suffer from very low precision, indicating a tendency to over-predict the positive class. Conversely, models like InceptionResNetV2 achieve high precision (0.8914) but at the expense of lower recall (0.5289). Our method, however, attains a strong balance, achieving a high precision of 0.8222 while maintaining a competitive recall of 0.7719.

To offer a more granular analysis, Figure 3 displays the confusion matrices for all compared methods. The heatmap for our model (bottom right) provides a clear visualization of its balanced performance. It correctly identified 1762 negative cases (TN) and 467 positive cases (TP). More importantly, the number of misdiagnoses (False Positives, FP=101) and missed diagnoses (False Negatives, FN=138) are both effectively suppressed. This contrasts sharply with models like InceptionResnetV2, which, despite having very few FPs (FP=59, indicating a low rate of misdiagnosing healthy cases), missed a significant number of positive cases (FN=283), posing a high risk of missed diagnosis. Our framework's ability to minimize both FN and FP demonstrates its robustness and clinical potential in handling imbalanced diagnostic data, achieving an optimal balance between identifying patients and avoiding false alarms.

To further assess the model's performance across all classification thresholds, we plotted the ROC and Precision-Recall (P-R) curves, as shown in Figure 4. In the ROC analysis (Figure 4a), our model achieves a superior AUC of 0.939, indicating its strong overall discriminative ability. More importantly, given the class imbalance of our dataset, the P-R curve (Figure 4b) provides a more insightful evaluation. Our model again leads with the highest Average Precision of 0.885. Its P-R curve is positioned consistently above all others, demonstrating a robust ability to maintain high precision even as recall increases. Both metrics confirm the comprehensive superiority of our proposed framework.

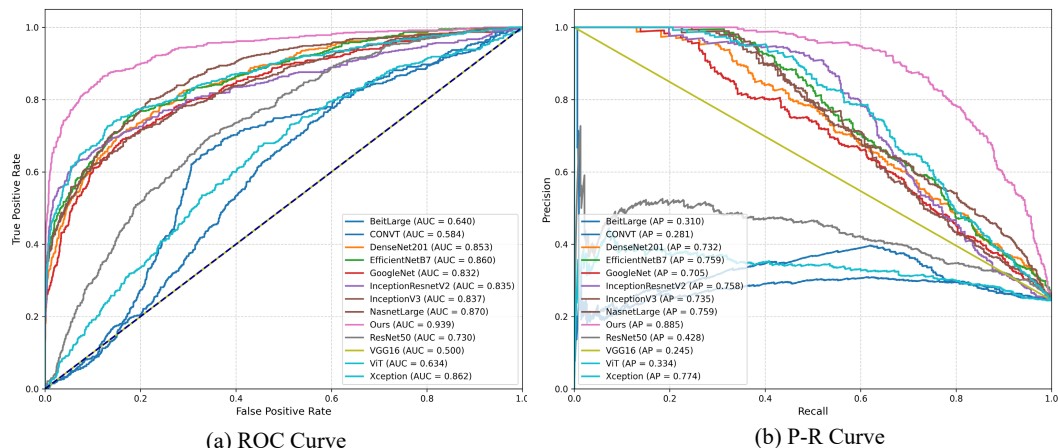

(a) ROC Curve                    (b) P-R Curve

Figure 4: (a) This chart displays the Receiver Operating Characteristic curves. The proposed model achieves the best performance with a leading Area Under the Curve (AUC) score of 0.939. This is notably higher than other models. (b) This chart displays the Precision-Recall curves. The proposed model again shows superior performance, attaining the highest Average Precision score of 0.885.

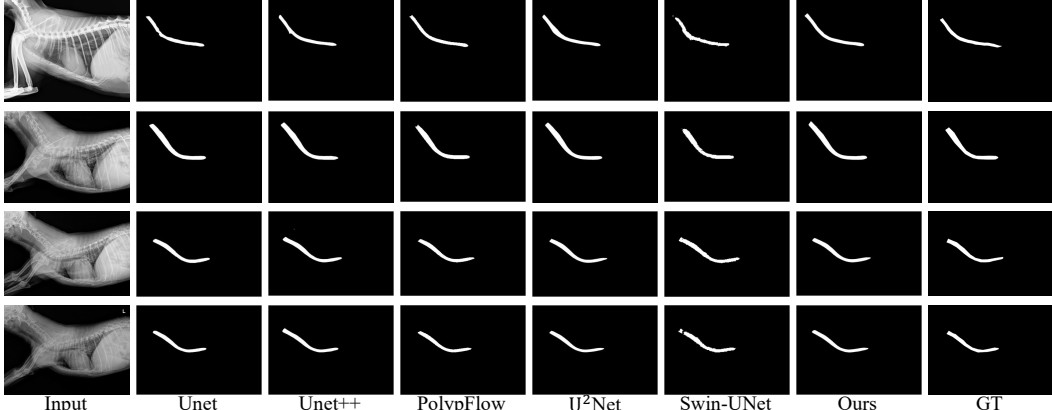

Input        Unet        Unet++        PolypFlow        U²Net        Swin-UNet        Ours        GT

Figure 5: **Qualitative Comparison of Unet-based Segmentation Results.** This figure presents a visual comparison of the segmentation performance of our proposed model against five Unet-based methods.

### 4.5 QUALITATIVE RESULTS

To visually substantiate our quantitative findings, we provide qualitative comparisons of the segmentation results. As shown in Figure 5, while U-Net-based models can capture the general shape of the target, our method produces cleaner boundaries and more accurate contours. More significant performance gaps are observed against other architectural families. For instance, the general-purpose model SAM (Figure 6) and Transformer-based models like Swin-Transformer (Figure 7) largely fail on this task, producing severely fragmented or noisy results. In contrast, our model robustly and accurately segments the target structure in all cases. These visualizations are in strong agreement with our superior quantitative metrics and demonstrate the practical effectiveness of our approach.

### 4.6 ABLATION STUDY

We conducted a series of ablation studies to validate the effectiveness of our framework's key components, with the results presented in Table 3.

Our segmentation ablation reveals that adding only VLM text guidance (b) degrades the baseline (a) performance. However, the Flow Matching module (c) is crucial for refining this raw guidance,

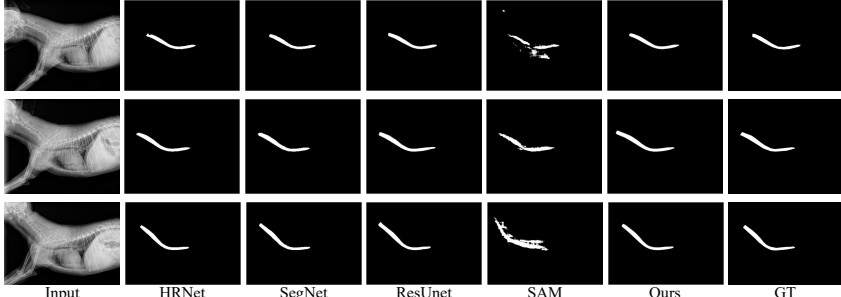

Figure 6: **Qualitative Comparison of Segmentation Results.** This figure presents a visual comparison of the segmentation performance of our proposed model against four methods.

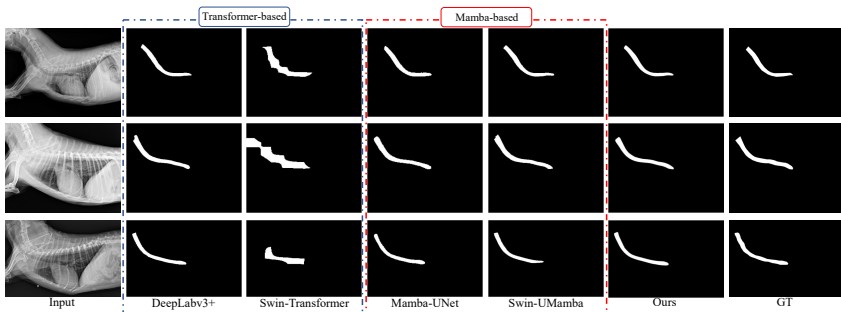

Figure 7: **Qualitative Comparison of Transformer-based and Mamba-based Segmentation Results.**

creating a synergistic effect that significantly surpasses the baseline with an mIoU of 0.8114. The value of segmentation for classification is clear: focusing the input on the segmented lesion improved the F1-score from 0.7209 (full image) to 0.7962, confirming a strong synergistic benefit.

Table 3: Ablation study of our proposed framework.

| Setting | Model Components | | | Seg. Performance | | Class. Performance | |
|---|---|---|---|---|---|---|---|
| | Text Guidance | Flow Matching | RMT Input (Purification) | mDice ↑ | mIoU ↑ | AUC ↑ | F1-Score ↑ |
| Experiment 1: Ablation on Segmentation Components | | | | | | | |
| (a) | × | × | – | 0.8830 | 0.7949 | – | – |
| (b) | ✓ | × | – | 0.8736 | 0.7051 | – | – |
| (c) | ✓ | ✓ | – | **0.8953** | **0.8114** | – | – |
| Experiment 2: Ablation on Classification Synergy | | | | | | | |
| (d) | ✓ | ✓ | Full Image | – | – | 0.9054 | 0.7209 |
| (e) | ✓ | ✓ | Focused Image | – | – | **0.9390** | **0.7962** |

## 5 CONCLUSION

In this work, we introduce a novel, interpretable framework for canine pneumothorax diagnosis and release the first accompanying public, pixel-level annotated dataset. Our method uniquely unifies VLM-guided Flow Matching for precise lesion localization with Random Matrix Theory (RMT) for diagnosis, reframing the task as the detection of statistical anomalies in purified pathological signals. This synergistic paradigm is proven to significantly outperform state-of-the-art models, offering a new path for developing trustworthy medical AI in data-scarce environments.

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
