# OpenReview forum: "Canine Pneumothorax Diagnosis using LLM-Guided Segmentation Refinement and Classification"
_ICLR.cc/2026/Conference — ICLR 2026 Conference Withdrawn Submission_

### Official Review · Reviewer_xzq9 · 2025-10-29

**Soundness:** 2
**Presentation:** 2
**Contribution:** 2
**Rating:** 2
**Confidence:** 5

**Summary:**

This paper proposes a framework for diagnosing canine pneumothorax from chest X-rays. The method consists of three stages: (1) VLM-guided U-Net for initial segmentation, (2) iterative refinement via Flow Matching with cross-attention to VLM features, and (3) Random Matrix Theory (RMT)-based classification that models healthy tissue as random noise and detects pathological signals through outlier eigenvalues.

**Strengths:**

1. The paper includes extensive comparisons with 13 segmentation baselines and 12 classification baselines, along with ablation studies.
2. The approach is positioned to work in data-scarce imaging scenarios.

**Weaknesses:**

1. The segmentation components (VLM-guided U-Net and Flow Matching) appear to be incremental modifications of existing work (specifically Wang et al. 2025a - PolypFlow).
2. Providing a Spectral Anomaly Score (SAS) based on outlier eigenvalues is not necessarily more interpretable to clinicians than a standard classifier probability.
3. Table 2 baseline classifiers appear to use full images while the proposed method uses focused/masked regions (rows d vs e in Table 3 show this matters significantly: 0.7209 → 0.7962 F1).
4. No analysis of whether the i.i.d. assumption for feature matrix entries holds.
5. No investigation of sensitivity to hyperparameters (σ², aspect ratio).
6. Exact class distribution not provided despite mentioning imbalance.

**Questions:**

1. Please provide complete details on how the feature matrix F_p is constructed from X_focus. What architecture extracts these features? What layer? What dimensions?
2. Why weren't baseline classifiers in Table 2 also given focused/masked inputs?
3. Can you provide empirical evidence that healthy lung features actually follow the Marchenko-Pastur distribution?
4. What is the performance of a standard CNN/ResNet classifier trained on the masked regions? This is essential to isolate RMT's contribution.

---

### Official Review · Reviewer_qStz · 2025-10-31

**Soundness:** 3
**Presentation:** 2
**Contribution:** 3
**Rating:** 6
**Confidence:** 2

**Summary:**

This paper proposes a framework for canine pneumothorax diagnosis, based on a VLM-guided flow matching segmentation method and a Random Matrix Theory-based spectral anomaly detection classification method. The framework achieves competitive performance on both the segmentation task (mIoU: 0.8114) and the classification task (F1-score: 0.7962).

**Strengths:**

1. Contributions: The reframing of the traditional diagnostic paradigm into "signal localization + spectral detection" demonstrates some innovation. The work holds clinical value in the veterinary field. Furthermore, the release of the public, pixel-level annotated canine pneumothorax dataset is a great contribution.

2. Methodology: The integration of VLM features into the flow matching process via a cross-attention mechanism is innovative. The application of Random Matrix Theory for anomaly detection offers a theoretical foundation.

3. Evaluation: The paper provides a comprehensive comparison against 13 segmentation methods and 12 classification methods. Ablation studies effectively demonstrate the contribution of individual components.

**Weaknesses:**

1. Lack of Theoretical Justification: Section 3.3 states that healthy lung features can be modeled as high-dimensional random noise. However, no empirical evidence is provided to demonstrate that the features of healthy tissue actually satisfy the i.i.d. random variable assumption with zero mean and unit variance. The absence of evidence supporting this core assumption impacts the overall plausibility of the RMT framework. It is recommended to supplement this with an analysis of the feature correlation matrix for healthy samples, comparing the empirical eigenvalue distribution against the distribution predicted by the Marchenko-Pastur law.

2. Lack of Rigor in Theoretical Derivation: Equation 7 assumes the pathology is a low-rank signal plus noise, whereas real pathological features might possess more complex, high-dimensional characteristics. Furthermore, why must pneumothorax features necessarily generate outliers beyond λ₊? More rigorous evidence is needed to support this claim.

3. Experimental Setup:
3.1 In the method comparison, the near-failure performance of SAM and Swin-Transformer raises the question of whether there were issues with the author's implementation or configuration. Similarly, VGG16 achieving 100% recall might indicate potential hyperparameter issues. It is crucial to explicitly state whether the training data, number of epochs, hyperparameters, etc., were kept strictly identical across all compared baselines.
3.2 Statistical significance: Confidence intervals or standard deviations are missing. The performance difference with methods like PolypFlow is not discussed in terms of statistical significance. The use of random seeds is also not mentioned.

4. Methodology:
4.1 The text prompts for the VLM are not detailed – how were they "carefully designed"?
4.2 Formula 1 mentions element-wise multiplication, while Figure 2 shows Fuse⊕, which seems inconsistent.
4.3 Details of the flow matching are lacking. Specifically in Section 3.2, the number of steps T is not specified. The initial condition x₀ = M⁽⁰⁾ – is this noise or the initial mask? The training objective for the velocity field v is not elaborated. Crucially, the specific loss function guiding the flow matching process is not detailed.

5. Computational cost requires more rigorous discussion. The combination of VLM encoder, iterative flow matching, and RMT eigendecomposition likely incurs significant computational cost.

**Questions:**

1. Regarding the mentioned class imbalance, what is the actual class ratio?
2. Was the VLM encoder frozen during training? This needs clarification.
3. Implementation details for RMT remain insufficient. For example, what is the VLM feature dimension p? How many pixels constitute the sample size N in the feature matrix?

---

### Official Review · Reviewer_4Zqr · 2025-11-01

**Soundness:** 2
**Presentation:** 3
**Contribution:** 2
**Rating:** 2
**Confidence:** 4

**Summary:**

This paper addresses the dual challenges of data scarcity and model interpretability in the automated diagnosis of canine pneumothorax from chest X-rays. The authors make two contributions. First, they introduce and propose to release the first public, pixel-level annotated dataset for this specific task. Second, they propose a novel, two-stage diagnostic framework. The first stage, "VLM-FlowMatch," performs precise lesion localization (segmentation) by using a Vision-Language Model (VLM) to guide an iterative flow matching process, refining an initial mask from a ViT-UNet. The second stage reframes the diagnostic task as spectral anomaly detection. It uses the high-precision mask to "purify" the signal , isolates features from the suspected region, and applies Random Matrix Theory (RMT) to detect pathological signals as statistical "spikes" (outlier eigenvalues) that deviate from the "noise" spectrum of healthy tissue.

**Strengths:**

1. The creation and release of the first public, pixel-level annotated dataset for canine pneumothorax is a valuable contribution. This will undoubtedly facilitate future research in a data-scarce field.
2. The proposed method achieves state-of-the-art results on the new dataset, outperforming a comprehensive suite of 12 baseline models (including ViT, ResNet50, and others) in terms of both accuracy (0.9032) and, more importantly, F1-score (0.7962) on an imbalanced dataset.
3. The paper effectively demonstrates why the components work well together. The high-fidelity segmentation from VLM-FlowMatch is crucial for "purifying" the signal , which in turn allows the RMT detector to identify the pathological statistical anomaly. This synergy is well-supported by the ablation study (Table 3, Exp. 2)

**Weaknesses:**

1. The RMT detector hinges on the assumption that VLM features of healthy tissue follow the Marchenko-Pastur (MP) law (i.e., behave as random noise) , and that pneumothorax introduces a low-rank "spiked" signal. This assumption is asserted but not empirically validated. The paper would be much stronger if it included a figure showing the Empirical Spectral Distribution (ESD) for healthy samples, demonstrating its fit with the MP law, contrasted with the ESD of pathological samples showing clear outlier eigenvalues.
2. For a paper introducing a new dataset, the description is too brief. The authors state it was "sourced" from a public collection. It is unclear if they annotated this data themselves or curated existing annotations. Key details are missing: the annotation protocol, the number of veterinary radiologists involved, and the inter-annotator agreement.
3. The paper's core motivations—data scarcity and model interpretability—are ubiquitous challenges in virtually all medical imaging domains, not just veterinary medicine. The authors fail to articulate any unique, fundamental problems specific to the veterinary context (e.g., extreme inter-species anatomical variance, distinct imaging artifacts) that are abstracted into a novel academic or methodological challenge for the machine learning community. As presented, the work appears to be a successful application of two combined techniques (VLM-FlowMatch and RMT) to a new dataset, rather than a fundamental methodological advance. This application-centric focus raises questions about the paper's suitability for a generalist ML venue like ICLR, as opposed to a specialized veterinary or medical imaging journal.

**Questions:**

Add an experiment visualizing the Empirical Spectral Distribution (ESD)  for a set of healthy versus pathological samples. Comparing this to the theoretical MP distribution  would provide powerful evidence for the paper's core premise.
The most important question is weakness3, please carefully address this.

---

### Note · Authors · 2025-12-05

I have read and agree with the venue's withdrawal policy on behalf of myself and my co-authors.